

# Identifying potentially marker symptoms of attention-deficit/hyperactivity disorder

Víctor B. Arias[1,2], Igor Esnaola[3] and Jairo Rodríguez-Medina[4]

[1] Department of Personality, Assessment and Psychological Treatment, University of Salamanca, Salamanca, Spain

[2] Institute on Community Integration (INICO), University of Salamanca, Spain

[3] Department of Developmental and Educational Psychology, University of the Basque Country, San Sebastian, Basque Country, Spain

[4] Department of Pedagogy, University of Valladolid, Valladolid, Spain

## ABSTRACT

**Background**. For the diagnosis of attention-deficit/hyperactivity disorder (ADHD), the Diagnostic and Statistical Manual of Mental Disorders (DSM-5) proposes that adherence to six symptoms in either group (inattention and hyperactivity/impulsivity) will lead to the diagnosis of one of three presentations of the disorder. Underlying this diagnostic algorithm is the assumption that the 18 symptoms have equal relevance for the diagnosis of ADHD, all are equally severe, and all have the same power to detect the presence of the disorder in all its degrees of severity, without considering the possibility of using marker symptoms. However, several studies have suggested that ADHD symptoms differ in both their power to discriminate the presence of the disorder and the degree of severity they represent. The aim of the present study was to replicate the results of previous research by evaluating the discriminative capacity and relative severity of ADHD symptoms, as well as to extend the investigation of this topic to Spanish-speaking Latin American samples.

**Methods**. The properties of ADHD symptoms rated by the parents of 474 Chilean children were analyzed. Symptom parameters were estimated using the graded response model.

**Results**. The results suggest that symptoms of ADHD differ substantially in both the accuracy with which they reflect the presence of the disorder, and their relative severity. Symptoms ''easily distracted by extraneous stimuli'' and ''have difficulty sustaining attention in tasks'' (inattention) and ''is on the go, acting as if driven by motor'' (hyperactivity/impulsivity) were the most informative, and those with relatively lower severity thresholds.

**Discussion**. The fact that symptoms differ substantially in the probability of being observed conditionally to the trait level suggests the need to refine the diagnostic process by weighting the severity of the symptom, and even to assess the possibility of defining ADHD marker symptoms, as has been done in other disorders.

Corresponding author
Víctor B. Arias, vbarias@usal.es, victor.arias@gmail.com

## INTRODUCTION

Attention deficit hyperactivity disorder (ADHD) is a neurodevelopmental disorder with a genetic basis, characterized by persistent symptoms of inattention (IN) and/or hyperactivity and impulsivity (HI), which are maladaptive and incoherent with regard to the child's level of development (*APA, 2013*). The prevalence of ADHD, according to the most recent edition of the Diagnostic and Statistical Manual of Mental Disorders (DSM-5), is between 3% and 7%, although these figures tend to present variations depending on the assessment instruments, the informants, the type of sample and other variables (*Barkley & Murphy, 2006*; *Bauermeister et al., 1990*; *Reid et al., 1998*).

For the diagnosis of ADHD, the DSM-5 requires the presence of at least six symptoms in either of the two groups, inattention (IN) and/or hyperactivity-impulsivity (HI), giving rise to three possible diagnoses: combined ADHD, ADHD with a predominance of attention deficit, or ADHD with a predominance of hyperactivity/impulsivity. To this diagnostic algorithm underlies the assumption that the 18 symptoms are equally relevant for the diagnosis of ADHD, all are equally severe, and all have the same reliability to detect the presence of the disorder in all its degrees of severity. Thus, any of the possible combinations of six or more symptoms will comply with the diagnostic criterion B, without considering the existence of marker symptoms, unlike other disorders such as major depression, whereby depressed mood and/or loss of interest are a necessary condition for diagnosis (*APA, 2013*). However, the application of this algorithm could lead to paradoxical situations such as, for example, the hypothetical case of a child diagnosed with predominantly inattentive ADHD who nevertheless does not show symptoms such as lack of sustained attention (symptom 2) or distractibility by irrelevant stimuli (symptom 8), indicators that are considered central for the assessment of the disorder (*Barkley, 2006*).

The dimensionality and internal structure of ADHD symptoms has been intensively researched in the last decades, in both the models proposed by the DSM and the International Statistical Classification of Diseases (ICD) (*Bauermeister et al., 2010*; *Willcut et al., 2012*) and alternative models that attempt to explain the high covariance among the totality of ADHD symptoms (e.g., *Burns et al., 2014*; *Arias, Ponce & Núñez, 2016*). Most of these studies have been based on factor analysis. Although factorial analysis can evaluate aspects related to the discriminative capacity of the symptom (e.g., through the examination of factor loadings), in general, its objective has been to investigate the general structure of the disorder and its relationship with other variables, not the properties of the particular symptoms.

The studies devoted to the evaluation of individual symptom properties have been much scarcer. The majority have consisted of applications of Item Response Theory (*Arias et al., 2016*; *Gomez, 2007*; *Gomez, 2008*; *Gomez, 2011*; *Gomez, 2012*; *Gomez, Vance & Gomez, 2011*; *Polanczyk et al., 2010*; *Purpura, Wilson & Lonigan, 2010*; *Li et al., 2016*; *Young et al., 2009*). Without exception, these studies have concluded that the symptoms of ADHD differ both in their ability to discriminate the presence of the disorder and in the degree of severity they represent. The results of these studies show variations, which is expected given the sample heterogeneity (e.g., different age ranges, sample types and informants)
and the use of different models (such as the graded response model, the two parameters logistic model, or the generalized partial credit model). However, these studies have attained a high consistency in regard to which symptoms are the most and least accurate for measuring ADHD in children. In the case of inattention, symptoms 2 (has difficulty sustaining attention) and 8 (is easily distracted by extraneous stimuli) have systematically been the most discriminative of the presence of the disorder; in contrast, symptoms 3 (does not seem to listen when spoken to directly) and, especially, 7 (loses things necessary for tasks or activities) have repeatedly been relatively less reliable than the rest. In the case of hyperactivity/impulsivity, the results have been less consistent, but there is a clear trend of symptoms 3 (runs and climbs in situations where it is inappropiate) and 4 (unable to play or engage in leisure activities quietly) as, respectively, the most and least discriminative among different studies. Regarding the severity of the symptom, it is observed (taking into account the beta parameter of the last threshold in the case of studies with polytomic items), a tendency of symptoms 8 of IN (easily distracted) and 6 of HI (talks excessively) to be placed in lower ranges of severity, and symptoms 7 of IN (loses things) and 7 of HI (blurts out answers before a question is complete) to represent the highest levels of severity. Given these results, attributing the same diagnostic weight to all ADHD symptoms is not compatible with the results of several empirical studies. On the contrary, such studies suggest the need to weigh the severity and discrimination of the symptom during the diagnostic process.

The aim of the present study was to replicate the results of previous studies by assessing the discriminative power and the relative severity of the symptoms of ADHD, under the hypothesis that (a) symptoms differ in the accuracy with which they reflect the presence of the disorder, and (b) symptoms are associated with different degrees of ADHD severity. To do so, we applied the graded response model (*Samejima, 1997*; *Samejima, 2010*) to the data collected from a large sample of Chilean children, rated by their parents using a scale constructed from the 18 ADHD symptoms of the DSM-5. Although the results of previous research present a clear trend, replication is necessary for the accumulation of empirical evidence that in the future may justify a change in the diagnostic criteria, especially in a population linguistically and culturally different from the english-speaking samples, on which most studies have focused. In addition, the IRT models allow a considerably deeper knowledge of the properties of the items (symptoms), which ideally contribute to the provision of accurate and valid instruments for the diagnosis, classification and evaluation of the effectiveness of the treatment. This need is especially important in ADHD, given the current doubts and controversies about its nature, prevalence, assessment and latent structure (*Bauermeister et al., 2010*; *Willcut et al., 2012*).

## MATERIALS AND METHODS

### Participants

The sample was composed of 474 children, aged between six and fifteen years (mean = 10.3; standard deviation = 2.3), of which 42% were male. The sample was obtained through the mediation of two schools from two regions of Chile, which voluntarily agreed to participate

in the study. Usual parent meetings were used to discuss the objectives of the research and invite the parents to participate. Parents (89% mothers) who agreed to participate were informed about how to respond to the questionnaire, which they completed with the direct support of the main researcher or one of the project collaborators. There were no cases with missing data due to the monitoring performed during data collection. In all cases, we obtained an informed consent for using the data in the research, with an acceptance rate of 100%. All procedures performed in this study were in accordance with the ethical standards of the institutional research committees (University of Talca, number 11140524) and the 1964 Helsinki Declaration and its later amendments or comparable ethical standards.

## Instrument

We used the ADHD subscale of the Child and Adolescent Behavior Inventory (CABI; *Burns et al., 2015*). This subscale is composed of 18 items corresponding to the ADHD symptoms proposed by the DSM-5 (inattention and hyperactivity/impulsivity). Each symptom was rated on a 6-point scale (i.e., *almost never (never or about once per month), seldom (about once per week), sometimes (several times per week), often (about once per day), very often (several times per day),* and *almost always (many times per day))*, choosing the option that best described the frequency of their children's usual behavior (i.e., such behaviors cannot be explained by the presence of behavioral problems or by the children's misunderstanding of tasks and instructions). Earlier studies with children from Chile and Spain provide support for the reliability and validity of scores for the ADHD-IN and ADHD-HI scales of the CABI (e.g., *Belmar et al., 2017*; *Burns et al., 2013*; *Sáez et al., 2018*).

## Data analysis

The data were analyzed with the IRTPRO 4.0 software (*Cai, Du Toit & Thissen, 2011*), using the graded response model (GRM; *Samejima, 1997*; *Samejima, 2010*). The GRM assumes, in addition to the usual IRT assumptions, that the categories to which the individual responds (or in which the individual is rated, as is in this case) can be ordered or hierarchized, as is the case, for example, of probability scales with summative estimates, or Likert-type scales. The model aims to obtain more information than available with only two response levels (e.g., "yes"—"no"), and in that sense, it is an extension of the two-parameter logistic model (2-PLM) to ordered polytomous categories.

The GRM specifies the probability of a person being rated with a category ik or higher as opposed to being rated with a lower category when the rating scale has at least three categories, and is expressed as

$$P^*_{ik}(\theta_j) = \frac{e^{D\alpha_i(\theta_j - \beta_{ik})}}{1 + e^{D\alpha_i(\theta_j - \beta_{ik})}} \tag{1}$$

$$P_{ik}(\theta_j) = P^*_{ik}(\theta_j) - P^*_{ik+1}(\theta_j) \tag{2}$$

where $k$ is the ordered response option; $P_{ik}(\theta_j)$ is the probability of responding to option $k$ of item $i$ with a latent trait level $\theta_j$; $P^*_{ik}(\theta_j)$ is the probability of responding to option $k$ or higher in item $i$ with a latent trait level $\theta_j$; $\theta_j$ is the latent trait level of the person; $\beta_{ik}$ is the location parameter of the alternative $k$ of item $i$; $\alpha_i$ is the discrimination parameter of item $i$, and $D$ is the constant 1.702.

Prior to the estimation of the GRM models, it was verified that the data complied with the necessary criteria of unidimensionality and local independence by means of a parallel analysis based on minimum rank factor analysis (*Timmerman & Lorenzo-Seva, 2011*) implemented in FACTOR 10.8 (*Lorenzo-Seva & Ferrando, 2013*), where the variance extracted in the first factor was compared with that obtained from 500 permutations of the sample values, and the inspection of the standardized LD-$\chi 2$ values of the matrix of expected and observed response frequencies to each item (LD-$\chi 2$ values greater than 10 suggest a substantial violation of local independence) (*Cai, Thissen & Du Toit, 2011*).

For the assessment of model fit, we examined the statistic M2 (*Maydeu-Olivares & Joe, 2006*), its associated root mean square error of approximation (RMSEA), and the differences between the response frequencies observed and expected by the model in each item by the inspection of the significance of S-$\chi 2$ (*Orlando & Thissen, 2000*). In the case of M2, non-significant values ($p > 0.01$) and associated RMSEA values close to zero ($<.08$; *Hu & Bentler, 1999*) are expected for a good model fit. In the case of S-$\chi 2$, good fit requires most items to have non-significant values ($p > 0.01$).

## RESULTS

### Unidimensionality and local independence

Figure 1 shows the result of the parallel analysis. For each sub-scale, the presence of a single clearly dominant factor can be observed, since only one eigen-value of each scale exceeds those obtained from random matrices (500 permutations). The LD-$\chi 2$ values associated with the expected frequencies against those values observed in both subscales were between .10 and 15.4 ($M = 5.1$, SD $= 3.04$). Two of the 42 contrasts were greater than 10, without observing clusters of extreme LD-$\chi 2$ values that could suggest the presence of substantive sources of non-modeled systematic error. As a result, the criteria of unidimensionality and local independence can be considered adequately complied with, and estimation of the GRM models proceeded.

### Fitting of the data to the GRM model

To assess the fit of the data to the GRM model, the $M 2$ and RMSEA statistics were examined for each of the scales. The results for both the IN scale ($M 2 = 1,529.9$, $df = 891$; $p = .0001$, RMSEA $=. 04$) and for the HI scale ($M 2 = 1,285$, $GL = 891$; $p = .0001$, RMSEA $= .03$) may indicate some lack of fit. However, the low associated RMSEA values suggest that this misfit may be due to a limited amount of unmodeled error (*Cai, Thissen & Du Toit, 2011*). Most of the items reached non-significant S-$\chi 2$ values ($p > .01$), in both sub-scales (six cases in IN and nine in HI).

### Estimation of the parameters

To estimate the item parameters $\alpha_i$ and $\beta_{ik}$, a marginal maximum likelihood method was used, given the sufficiently large sample size. Figure 2 shows, as an example, the response category curves corresponding to item 2 (sustained attention) of the IN scale. The abscissa represents the latent variable $\theta$ ($M = 0$; SD $= 1$). For each item, six curves are drawn, each of which represents the probability, represented on the ordinate axis, of being located
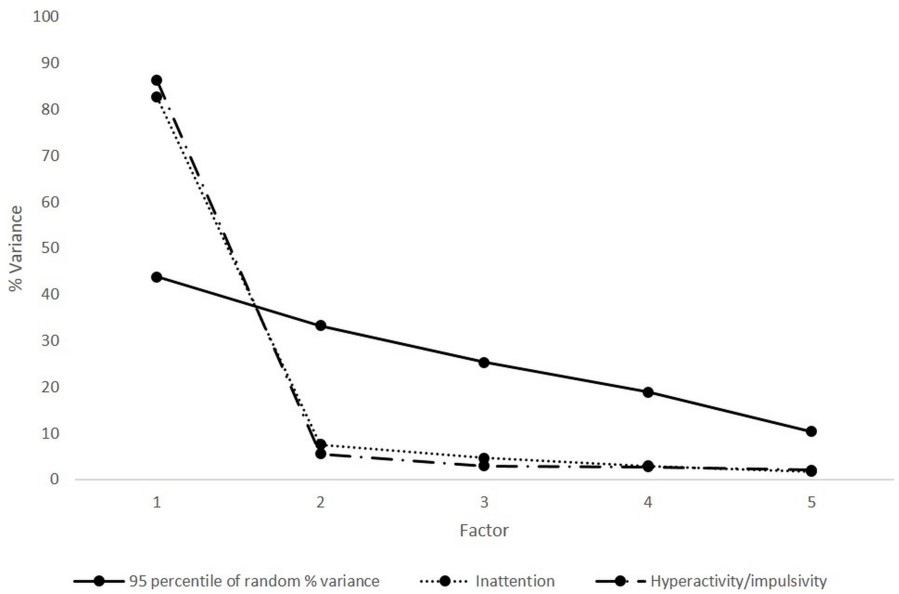

**Figure 1  Parallel analysis results.**

in each of the response categories conditionally to the trait level. Thus, in the symptom discussed, for the most likely response to be "many times a day" (highest category), the child must present a very high level of inattention (approximately 2 standard deviations or more above the mean). On the other hand, the most likely score of a child near the mean (Theta = 0) would be "once a week" (category 2).

Table 1 shows the values of the parameters $\alpha_i$ and $\beta_{ik}$. Since, according to *Baker (2001)*, $\alpha_i$ values from 0.01 to 0.24 are very low, from 0.25 to 0.64 are low, from 0.65 to 1.34 moderate, from 1.35 to 1.69 high, and more than 1.7 are very high, we conclude that, in absolute terms, the discrimination parameters have been very high (>1.70). In relative terms, we observe that in the IN scale the most discriminative items have been 2 (sustained attention, $\alpha_i = 3.73$) and 8 (easily distracted, $\alpha_i = 3.44$). On the contrary, while maintaining a reasonable level, symptom 7 (loses things, $\alpha_i = 1.73$) has presented substantially less discriminative power than the rest, so that its relative contribution to the test information has been lower. In the case of the HI scale, the three most discriminative symptoms were 5 (driven by a motor/on the go, $\alpha_i = 3.62$) 2 (leaves seat, $\alpha_i = 2.96$) and 7 (blurts answers, $\alpha_i = 2.94$). No symptom has been clearly less discriminative than the others.

Figure 3 shows the cumulative information functions of the items, where the relative contribution of each symptom to the total information of the test is illustrated (the upper curve—dotted line—simultaneously represents the cumulative information of the last item and the information function of the complete test; the items are sorted from bottom (item 1) to top (item 9) in the same order as in Table 1). As shown in Fig. 3A (IN sub-scale), items 2 (sustaining attention) and 8 (easily distracted) contribute more information to the test, while item 7 (loses things) contributes little information relative to the other symptoms. The complete test provides the largest amount of information ranging between

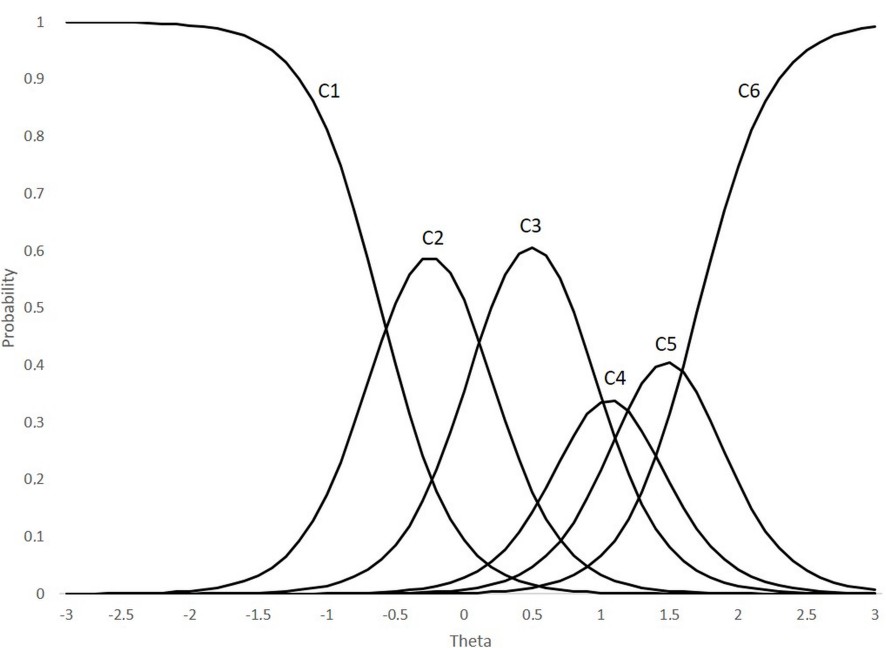

**Figure 2  Curves of response categories (item 2, inattention sub-scale).**

approximately −.5 and 2 standard deviations, producing the peak of maximum information in high areas of the latent variable (between 1 and 2 standard deviations above the mean). Figure 3B shows the items on the hyperactivity-impulsivity sub-scale. Clearly, item 5 (on the go/driven by a motor) provides the most information to the test set, and no symptoms are observed that provide substantially less information than the rest. The test acquires the maximum informative power in a range between the mean and +1.7 standard deviations.

Regarding the parameters $\beta_{ik}$ (Table 1), the values range from approximately −1 SD to +2 SD. As seen, the symptoms differ substantially in the degree of severity they represent. This result is illustrated in Fig. 4. Figure 4A shows the characteristic curves of the more and less severe inattention items (assuming category 4—"very often, several times a day"—as a threshold to consider the potential adhesion to the symptom). In the least severe item (IN8, easily distracted), an observed score of 4 is expected in children with a level of inattention approximately 1.4 standard deviations above the mean. On the other hand, for the most severe item (IN1, fails to give close attention), the same observed score is expected for substantially higher levels of inattention (2.2 standard deviations above the mean). The hyperactivity scale shows similar results (see Fig. 4B), with differences close to one standard deviation of severity between the most severe (HI3, runs and climbs $\theta = 1.3$) and least severe (HI7, talks excessively, $\theta = 2.1$).

## DISCUSSION

In the present study, the graded response model has been applied to a sample of Chilean children rated by their parents in the 18 symptoms of ADHD proposed by the DSM-5. The

**Table 1 Graded response model parameters.**

| Symptom | Content | a (SE) | Parameters | | | | |
|---|---|---|---|---|---|---|---|
| | | | $b_1$ | $b_2$ | $b_3$ | $b_4$ | $b_5$ |
| **IN sub-scale** | | | | | | | |
| IN 1 | Close attention | 2.34 (.18) | −.91 | .20 | 1.20 | 1.84 | 2.16 |
| IN 2 | Sustaining attention | 3.73 (.30) | −.61 | .12 | .87 | 1.25 | 1.71 |
| IN 3 | Listen | 2.90 (.22) | −.45 | .32 | 1.06 | 1.51 | 2.00 |
| IN 4 | Follow through | 2.98 (.24) | −.13 | .59 | 1.27 | 1.66 | 2.20 |
| IN 5 | Organizational skills | 2.72 (.22) | −.30 | .48 | 1.32 | 1.65 | 2.29 |
| IN 6 | Avoid tasks | 2.95 (.23) | −.33 | .44 | 1.04 | 1.57 | 2.03 |
| IN 7 | Loses things | 1.73 (.14) | −.66 | .32 | 1.15 | 1.72 | 2.19 |
| IN 8 | Easily distracted | 3.44 (.18) | −.75 | .08 | .75 | 1.13 | 1.51 |
| IN 9 | Forgetful | 2.72 (.18) | −.52 | .35 | .91 | 1.45 | 1.92 |
| **HI sub-scale** | | | | | | | |
| HI 1 | Fidgets/squirms | 2.56 (.22) | .05 | .73 | 1.27 | 1.60 | 1.91 |
| HI 2 | Leaves seat | 2.96 (.25) | .09 | .78 | 1.38 | 1.78 | 2.12 |
| HI 3 | Runs/climbs | 2.74 (.25) | .43 | .98 | 1.63 | 2.02 | 2.26 |
| HI 4 | Playing quietly | 2.60 (.22) | .02 | .73 | 1.33 | 1.56 | 1.93 |
| HI 5 | On the go/driven by a motor | 3.62 (.33) | .09 | .65 | 1.13 | 1.39 | 1.76 |
| HI 6 | Talks excessively | 2.62 (.21) | −.65 | .10 | .71 | 1.07 | 1.29 |
| HI 7 | Blurts answers | 2.94 (.24) | −.45 | .32 | .97 | 1.37 | 1.75 |
| HI 8 | Awaiting turn | 2.73 (.23) | −.01 | .68 | 1.23 | 1.56 | 1.91 |
| HI 9 | Interrupts/intrudes | 2.52 (.20) | −.23 | .55 | 1.21 | 1.54 | 1.87 |

**Notes.**

IN, Inattention; HI, Hiperactivity/impulsivity; SE, Standard Error.

results suggest that, in the general population, the symptoms present adequate psychometric properties and are configured as a semi-trait, where most of the information is located at high levels of the latent variable (an expected result given the clinical nature of the scale; *Reise & Waller, 2009*).

However, the results suggest that the symptoms of ADHD differ substantially, both in the accuracy with which they reflect the presence of the disorder and in its relative severity. Consistent with previous studies (e.g., *Gomez, 2008*), symptoms 8 (easily distracted) and 2 (sustaining attention) of the inattention sub-scale have been the most informative, yet at the same time, reflect less severity of the disorder. Given these results and their relative stability between different studies, these symptoms are good marker candidates of the disorder. In the case of the hyperactivity-impulsivity sub-scale, the symptoms have been more homogeneous in regard to their discriminatory power; nevertheless, symptom 7 (on the go/driven by a motor) seems to be the clearest candidate as a marker symptom, given its high information power and low severity, results also observed in previous studies (e.g., *Arias et al., 2016*).

On the other hand, symptom 7 (loses things) of the inattention sub-scale has presented low discrimination values relative to the rest of symptoms, as well as poorer information and lower reliability at all levels of the latent trait. Our results are compatible with previous

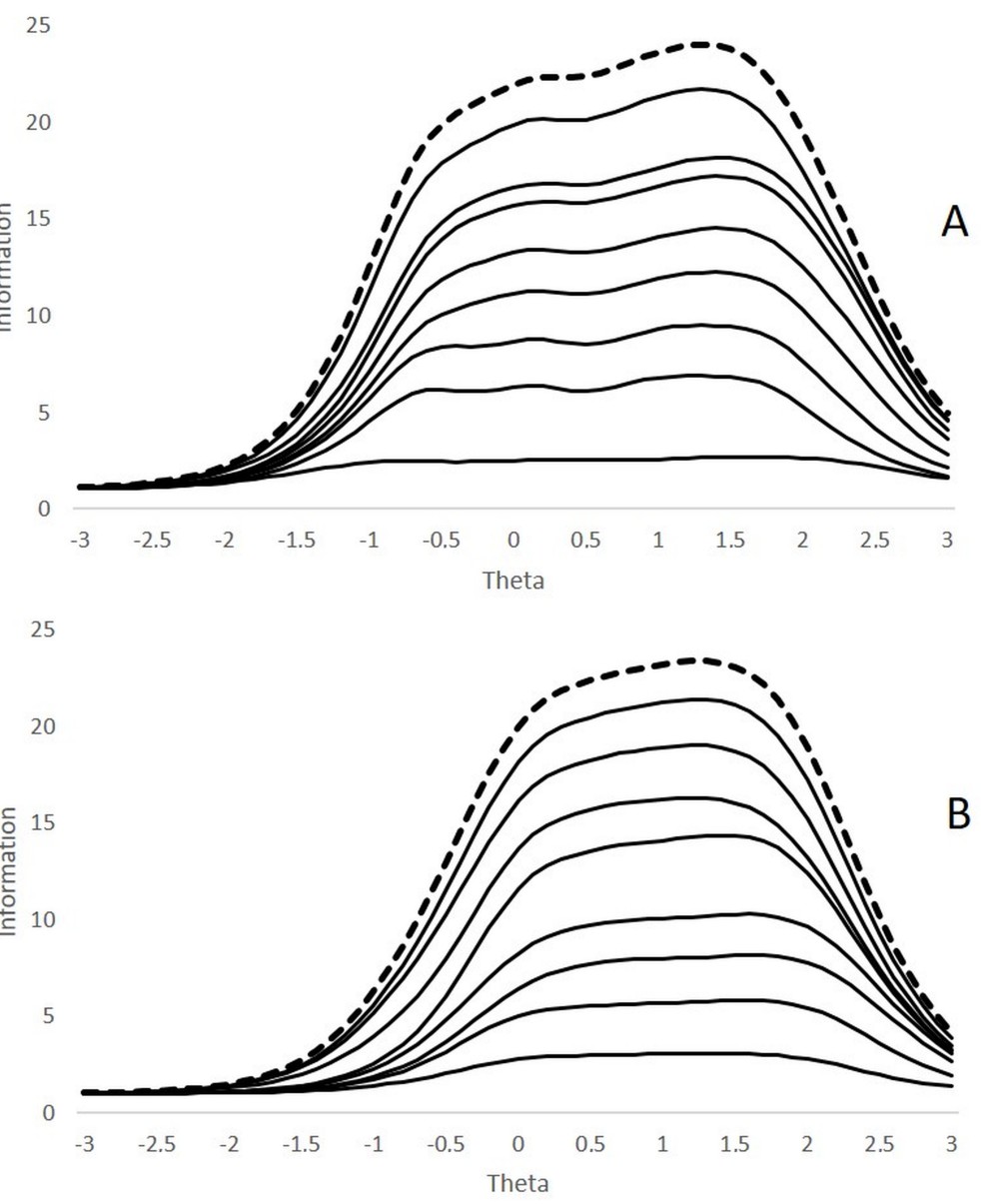

**Figure 3 Cumulative information functions of the items.** (A) Inattention items. (B) Hyperactivity-impulsivity items. The cumulative functions are sorted from bottom to top in the same order as shown in Table 1 (i.e., IN1 to IN9 and HI1 to HI9). The dotted line represents both the cumulative information function of the last item and the complete test information function.

evidence of its reduced informative capacity (*Gomez, 2008*). This finding suggests the need to either reduce the list of symptoms to those with greater discriminative power, resulting in a more parsimonious diagnostic process, or, as *Gomez (2008)* suggests, to rewrite the description of the symptom so that it captures the content to be evaluated with greater precision.

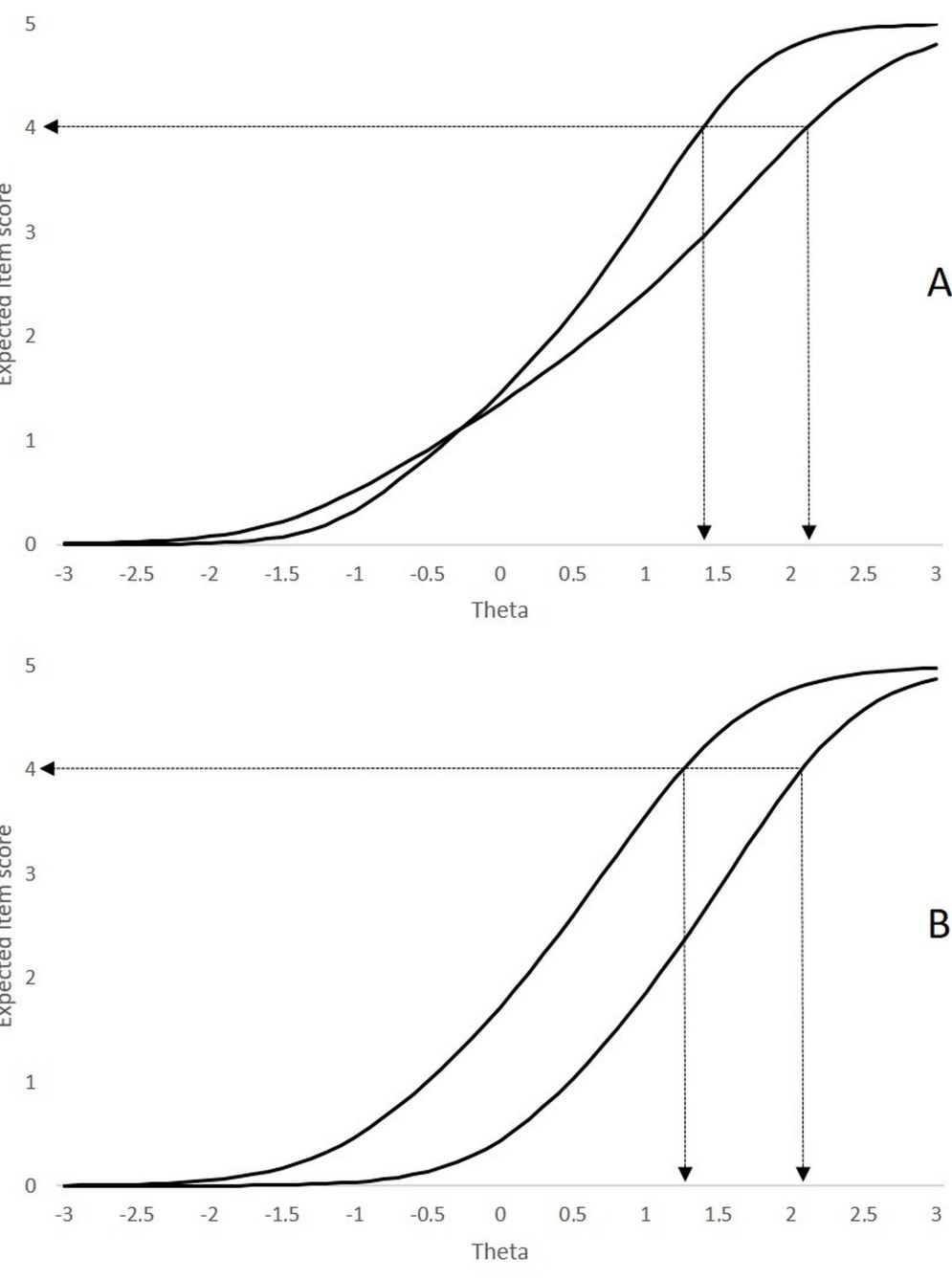

**Figure 4 Characteristic curves of the most and least severe symptoms.** (A) Characteristic curves of the most and least severe inattention symptoms (IN1 -fails to give close attention- and IN8 -easily distracted-, respectively). (B) Characteristic curves of most and least severe hyperactivity-impulsivity symptoms (HI3 -run and climbs-, and HI7 -talks excessively-, respectively).

In summary, the differences between symptoms in severity and power of discrimination suggest the need to identify which symptoms are more relevant than others for the diagnosis of the disorder and its subtypes (*Young et al., 2009*). Regarding the clinical assessment of ADHD, these results are in line with the hypothesis that ADHD is organized along a generalized continuum whose upper end can be understood as the clinical manifestation of the disorder. The fact that the symptoms differ substantially in the probability of being observed conditionally to the trait suggests the need to refine the diagnostic process by weighing the severity of the symptom, and even to assess the possibility of defining ADHD markers, as has been done in other disorders such as depression. From the above, it follows that perhaps the diagnostic algorithm proposed by the DSM, based on the categorization into subtypes or presentations by unweighted counting of endorsed symptoms (*APA, 2013*), is not the most appropriate diagnostic procedure. Although we have not compared different latent structures of the disorder, our results are compatible with previous studies that have used taxometric analysis or mixture models (*Marcus & Barry, 2011*; *Haslam et al., 2006*; *Lubke et al., 2009*), according to which the disorder could be considered as a spectrum of inattention and hyperactivity/impulsivity symptoms, where the presence or absence of the disorder is verified by the severity with which a subject presents the symptoms of IN and HI. Clearly, moving to a diagnostic criterion based on cut-off points along a weighted continuum of severity would require a major research effort that would provide strongly supported diagnostic decision norms. However, an approach like the one described would replace the current universal but categorical, inflexible and largely arbitrary cut-off point (6 symptoms) with norms adapted to aspects such as the development stage or the social and cultural context, thus allowing a much more reliable and flexible measurement of the disorder. In terms of treatment, moving from a categorical diagnosis (presence or absence of ADHD) to one based on the assessment of severity would facilitate the design of evidence-based interventions aimed at reducing the severity of the behavioral manifestations of the disorder (*Lubke et al., 2009*).

Finally, for a proper interpretation of the results, it is necessary to bear in mind some of the limitations of this study, which in turn would suggest possible future research. First, sample was non-probabilistic, so the generability of results to the population is limited. Second, the psychometric properties of the items were based on parental ratings; it would have been desirable to compare them with the teachers' scores, given that similar studies have shown certain differences in the usefulness of the symptoms by type of informant (e.g., *Gomez, 2008*). Third, it would have been desirable to use clinical samples in order to estimate the functioning of the items and the test at highest levels of severity, although previous results suggest that children in the general and clinical population can be assessed on the same continuum (*Li et al., 2016*).

## CONCLUSIONS

ADHD symptoms differ substantially in both the accuracy with which they reflect the presence of the disorder, and its relative severity. This suggests the need to refine the diagnostic process by weighting the severity of each symptom, and even to assess the

possibility of defining ADHD marker symptoms, as has been done in other disorders. These results also suggest that ADHD would be better assessed as a spectrum of symptoms that differ in severity along a continuum, than by categorizing subjects according to a fixed threshold of endorsed symptoms.

### Funding

Support was provided by Fondo Nacional de Desarrollo Científico y Tecnológico (Fondecyt) de Iniciación No. 11140524 (http://www.conicyt.cl/fondecyt/). The funders had no role in study design, data collection and analysis, decision to publish, or preparation of the manuscript.

### Grant Disclosures

The following grant information was disclosed by the authors:
Fondo Nacional de Desarrollo Científico y Tecnológico (Fondecyt) de Iniciación: 11140524.

### Competing Interests

The authors declare there are no competing interests.

### Author Contributions

- Víctor B. Arias conceived and designed the experiments, performed the experiments, analyzed the data, prepared figures and/or tables, authored or reviewed drafts of the paper, approved the final draft.
- Igor Esnaola and Jairo Rodríguez-Medina conceived and designed the experiments, contributed reagents/materials/analysis tools, prepared figures and/or tables, authored or reviewed drafts of the paper, approved the final draft.

### Human Ethics

The following information was supplied relating to ethical approvals (i.e., approving body and any reference numbers):
The University of Talca granted Ethical approval to carry out the study within its facilities (Ethical Application Ref: 11140524).

### Data Availability

The raw data are provided in a Supplemental File.

### Supplemental Information

Supplemental information for this article can be found online at http://dx.doi.org/10.7717/peerj.4820#supplemental-information.

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
