# Peer review of "Identifying potentially marker symptoms of attention-deficit/hyperactivity disorder"

_PeerJ, doi:10.7717/peerj.4820_

## Round 0.1 · original submission · Minor Revisions

I now have received two reviewers' comments. Although both reviewers expressed their interest in your study, several aspects of this manuscript should be revised to improve its clarity. Their observations are presented with clarity so I'll not risk confusing matters by belaboring or reiterating their comments. While I might quibble with the occasional point, I note that I regard the reviewers' opinions as substantive and well-informed. I believe that all of the highlighted reservations require contemplation and appropriate attention in revising the document if it is to contribute appropriately to Peerj and the extant literature. Please revise or refute according to the two reviewers' comments and provide a point by point reply in addition to the revised manuscript.

Tsung-Min Hung, Ph.D.
PeerJ editor
Distinguished professor
Department of Physical Education
National Taiwan Normal University

Reviewer 1 ·

Basic reporting

This study is well written, with clear and proper language throughout the manuscript. In both the introduction and the discussion sections the authors cite the relevant literature to properly frame the research problem and the study’s conclusions. The structure of the paper conforms to PeerJ’s standards, with the exception of the “Discussion” heading in the abstract, which does not begin on a new paragraph. All the figures and tables are relevant, of high quality, and are properly labeled and described. The authors supplied the raw data and the items in the file are properly labeled and coded.

Experimental design

The current study consists of an original primary research that falls within the scope of the PeerJ journal. The research question is well defined and relevant. The authors clearly state how this research fills a gap in the available literature. The research was performed to a high technical and ethical standard. The methods were described with sufficient detail to allow for replication, except for the parallel analyses, as commented on point #1.

Validity of the findings

The rationale and benefit to the literature was clearly stated by the authors. The data collected was robust and the analyses performed were appropriate to the research question. The conclusions were well stated, linked to the original research question and limited to the obtained results (with the exception of those commented on point #7).

Additional comments

The present work aims to replicate some relevant findings from the attention deficit hyperactivity disorder (ADHD) literature with a Spanish-speaking Chilean sample. Specifically, the study is concerned with assessing the discriminative power and relative severity of the symptoms of ADHD, as measured by the inattention and hyperactivity scales of the Child and Adolescent Behavior Inventory (CABI). The authors argue that considering the different psychometric properties of the item scores, as opposed to the current criteria outlined in the fifth edition of the Diagnostic and Statistical Manual of Mental Disorders (DSM-5) where all indicators of the disorder are given equal weight, might enhance the accuracy of the ADHD diagnosis. To this end, the authors employ Samejima’s graded response item response model to evaluate the discriminative power and relative severity of the ADHD symptoms, and provide evidence that indeed the properties of the item scores have substantial variation, in particular for the inattention scale. Finally, the authors suggest that the diagnosis process of ADHD could be improved by weighting the severity of each symptom, and by considering the possibility of using the most informative symptoms as markers.

All of my concerns about the paper are very minor and easy for the authors to address. I will list them next.

1) In the data analysis subsection line 155 the authors state that they used parallel analysis to evaluate the unidimensionality of the scales used in this study. However, they don’t give enough information regarding how they implemented the method to allow for replication. In this regard, the authors should inform of: a) the software/code they used; b) the criteria used to summarize the random eigenvalues (e.g., the mean, the 95th percentile, etc.); and c) the procedure used to generate the random variates (e.g., random column permutations of the empirical data, data generated from a multivariate distribution, etc.). Further, I believe that the information regarding the use of exploratory factor analysis to obtain the parallel analysis eigenvalues is erroneous, as the random eigenvalues have a mean of 1, which indicates that these are the eigenvalues from the full correlation matrix (i.e., the principal component eigenvalues).

2) In lines 165 and 166-167 the authors propose a Type I error rate of 0.01. Why 0.01 and not the standard 0.05?

3) In line 165 the authors should specify what they consider as RMSEA values close to zero, including the appropriate references.

4) As all the discrimination parameters shown in Table 1 are greater than 1.70, they all fall in the ‘very high’ range; in line 202 it says that they were ‘high’ or ‘very high’.

5) According to Table 1, the most discriminative item for the HI subscale was item 5 (it says item 6 in line 207). Additionally, in the table it says that the ‘a’ parameter was 3.62, while in line 207 it says 3.78. Also, the second and third most discriminative items have practically the same parameter estimate (2.96 and 2.94) so it may make more sense to mention them both in line 208.

6) The values of 1.13 and 1.84 given in lines 230 and 233 do not match the position of the lines in Figure 4. For example, for the most severe item a score of 4 touches the horizontal axis at a value higher than 2.

7) Although the assertion in lines 297 to 299 is supported by the literature (e.g., Haslam et al., 2006; Marcus & Barry, 2011), it cannot be derived from the analyses of the current study. To do that, the authors would need to perform a formal taxometric analysis. So, I suggest that the authors change the wording on this sentence to reflect that this is their view based on the literature and not on the results of the current study.

Reviewer 2 ·

Basic reporting

The quality of this study is good, with clear and professional English.

Experimental design

All is good, except for the method, as commented on point #1.

Validity of the findings

The findings were generally valid. My main concern here relates to the data collected was based on parental rating. I raise my comment on point #2.

Additional comments

The purpose of this study was to replicate the results of previous research by evaluating the discriminative capacity and relative severity of ADHD symptoms. The result suggests that the diagnosis process of ADHD could be improve by weighting the severity of the symptom and even to assess the possibility of defining ADHD marker symptoms.
My main concerns about this paper are list below:

1) You chose 6-15 years old students as your sample, which includes children, preadolescent children, and adolescent. The age range is wide and might affect the result. Have you consider this issue and use the adequate statistical method to eliminate age difference?

2) The psychometric properties of the items were based on parental rating, as you mentioned it in the research limitation. I believe that teacher’s rating might be an important data in your study. Since you didn’t collect these data, I would suggest authors can provide the reason.

3) Base on literature and background knowledge author provide in introduction. I would suggest author can provide hypothesis in the introduction. And see if the result meet your expectation or not.

---

## Round 0.2 · accepted · Accept

I have now received two reviewers’ comment and both reviewers were satisfied with your reply and revisions from previous comments. You and your coauthors have my congratulations. Thank you for choosing PeerJ as a venue for publishing your research work and I look forward to receiving more of your work in the future.

Tsung-Min Hung, PhD., FNAK
PeerJ editor
Research chair professor,
Department of Physical Education,
National Taiwan Normal University

# Reviewer 1 ·

Basic reporting

The basic reporting in this article meets the standards of PeerJ.

Experimental design

The research design of this study meets the standards of PeerJ.

Validity of the findings

The validity of the findings derived from this study meets the standards of PeerJ.

Additional comments

The authors have satisfied all my concerns. I therefore recommend this article for publication.

Reviewer 2 ·

Basic reporting

No Comments.

Experimental design

The authors have appropriately addressed previous concerns that were pointed out.

Validity of the findings

The authors have appropriately addressed previous concerns that were pointed out.

Additional comments

The authors have completely addressed the questions provided by the reviewer and have completely revised the manuscript. The reviewer considers this manuscript now is in good shape and has no further questions/concerns on this manuscript. Congratulations to the authors for their hard work and the effort they have made.